# Sex-specific transcriptomic responses to changes in the nutritional environment

M Florencia Camus[1]*, Matthew DW Piper[2], Max Reuter[1]

[1]Research Department of Genetics, Evolution and Environment, University College London, London, United Kingdom; [2]School of Biological Sciences, Monash University, Melbourne, Australia

**Abstract** Males and females typically pursue divergent reproductive strategies and accordingly require different dietary compositions to maximise their fitness. Here we move from identifying sex-specific optimal diets to understanding the molecular mechanisms that underlie male and female responses to dietary variation in *Drosophila melanogaster*. We examine male and female gene expression on male-optimal (carbohydrate-rich) and female-optimal (protein-rich) diets. We find that the sexes share a large core of metabolic genes that are concordantly regulated in response to dietary composition. However, we also observe smaller sets of genes with divergent and opposing regulation, most notably in reproductive genes which are over-expressed on each sex's optimal diet. Our results suggest that nutrient sensing output emanating from a shared metabolic machinery are reversed in males and females, leading to opposing diet-dependent regulation of reproduction in males and females. Further analysis and experiments suggest that this reverse regulation occurs within the IIS/TOR network.

DOI: https://doi.org/10.7554/eLife.47262.001

## Introduction

Sex differences in life history, behaviour and physiology are pervasive in nature. These differences arise mainly from the divergent reproductive strategies between the sexes that are rooted in anisogamy (*Chapman, 2006*). Typically, males produce large numbers of small, cheap gametes and evolve traits that facilitate the acquisition of mates and the increase of fertilisation success. Females, on the other hand, produce fewer, energetically costlier gametes and tend to evolve traits that optimise rates of converting resources into offspring (*Trivers and Campbell, 1972*). Given these fundamental differences between male and female reproductive investments, one of the key areas of divergence between the sexes concerns physiology, metabolism and responses to diet (*Jensen et al., 2015*).

Studies in insect species (*Jensen et al., 2015*; *Reddiex et al., 2013*; *Maklakov et al., 2008*; *Maklakov et al., 2009*; *Simpson and Raubenheimer, 2011*) have shown that the two sexes require different diets to maximise fitness. Female fitness is typically maximised on a high concentration of protein, which fulfils the demands of producing and provisioning eggs. Males, in contrast, achieve optimal fitness with a diet consisting of more carbohydrate, which can fuel activities such as locating and attracting mates. Work on nutritional choices has shown that individuals tailor their diet in line with their physiological needs. In insects, females overall prefer diets with higher protein content, whereas males chose a more carbohydrate-rich diet (*Lee et al., 2008*; *Corrales-Carvajal et al., 2016*). These choices are further adapted to reflect the individual's current condition and reproductive investment (*Corrales-Carvajal et al., 2016*; *Ribeiro and Dickson, 2010*). For example, *Camus et al. (2018)* found that the female preference for protein in fruit flies was significantly higher in mated females (who require resources to produce eggs) than virgins, while the preferences of males (who start producing sperm before reaching sexual maturity) did not significantly differ between mated and virgin flies.

*For correspondence:
f.camus@ucl.ac.uk

Competing interests: The authors declare that no competing interests exist.

**eLife digest** "You are what you eat" is a popular saying that can often make scientific sense. Everything an animal eats gets broken down into smaller molecules that fuel the many biological processes required to survive, move and reproduce. However, the food that the sexes need to maximize their fertility may not be exactly the same, as males make lots of small, mobile sperm cells while females create a small number of large eggs. In fruit flies for example, females benefit most from foods that contain lots of protein, while males are more fertile when they eat foods that are rich in carbohydrates. However, it remained unclear how these differences have evolved.

Here, Camus et al. examine the genes that are active in male and female fruit flies which eat a diet rich in either carbohydrates or in proteins. Their experiments showed that both sexes share a large collection of genes which respond to the two diets in the same way. However, the type of food had opposite effects on the activity of certain genes involved in male and female reproduction. When the fruit flies had a protein-rich diet, for example, genes that promoted reproduction got turned on in females, but switched off in males. The opposite pattern was observed when the insects were exposed to carbohydrate-rich diets. Further analyses suggested that these different responses might be linked to a molecular network called IIS/TOR, which is a specific cascade of reactions that responds to nutrient availability.

The findings of Camus et al. suggest that male and female flies produce different signals in reaction to food, which helps them to reproduce when they are able to meet their particular nutritional needs. Armed with a better understanding of the fundamental differences between the sexes, it may be possible to improve research into human health and animal keeping.
DOI: https://doi.org/10.7554/eLife.47262.002

But individuals not only choose diets to suit their needs where possible, they also adapt their physiology and reproductive investment in response to the quality and quantity of nutrition available. This has been studied extensively using experiments that either alter the macronutrients composition (carbohydrates vs. protein) of the diet while keeping the overall caloric intent constant, or by manipulating the overall nutrient content of the food—dietary restriction (DR). These studies have shown that a wide range of life history traits respond to changes in both the composition of the food (*Simpson and Raubenheimer, 2011*; *Moatt et al., 2019*; *Solon-Biet et al., 2014*) and the quantity of nutrients supplied (*Piper et al., 2005*; *Regan et al., 2016*; *Piper and Partridge, 2018*). For example, DR typically causes an extension of lifespan at the cost of reduced reproduction (*Partridge et al., 2005*), and a similar response can be triggered by a shift from protein to carbohydrates in the diet (*Solon-Biet et al., 2014*).

Although most studies manipulating diet have concentrated on females only, those including both sexes suggest that DR responses are broadly similar in males and females—despite their large differences in optimal diet. In fruit flies, DR extends lifespan in both sexes (*Magwere et al., 2004*; *Zajitschek et al., 2014*; *Zajitschek et al., 2013*), even though the observed increase in longevity appears smaller in males than females and the degree of DR that maximises lifespan can differ between the sexes (*Magwere et al., 2004*). Qualitatively similar results have been obtained when manipulating the macronutrient composition of the diet. Studying field crickets, (*Maklakov et al., 2008*) found that shifting the dietary balance away from protein and towards carbohydrates increased lifespan in both sexes, even though the effect of nutrients on reproductive investment differed between the sexes (*Maklakov et al., 2008*). These quantitative sex differences in dietary lifespan effects can at least in part be attributed to sex-biased responses in individual tissues. Thus, Regan and co-workers showed that *D. melanogaster* males in which the gut had been genetically feminised had DR responses more similar to those of females (*Regan et al., 2016*).

The contrast between large differences in optimal diet but similar responses to diet manipulation raises the question of how males and females differ in their diet-dependent regulation of metabolism and reproductive allocation. Due to the predominant focus on female responses to nutrition, we currently know relatively little about the degree to which regulation is shared or differs between the sexes (*Hoedjes et al., 2017*), in particular at the molecular level. Work in females has shown that nutrient-sensing pathways play a key role in the observed DR phenotype (*Clancy et al., 2002*;

*Slack et al., 2011*; *Zandveld et al., 2017*; *Emran et al., 2014*; *Bjedov et al., 2010*). Specifically, two evolutionarily conserved signalling pathways—insulin/insulin-like growth factor 1 (IIS) and Target of Rapamycin (TOR)—are thought to regulate longevity in a diet-dependent way (*Hoedjes et al., 2017*; *Alic and Partridge, 2011*; *Gallinetti et al., 2013*). Recent transcriptomic work in female *D. melanogaster* has further shown that DR and rapamycin treatment (which inhibits TORC1 activity) elicit similar changes in gene expression (*Dobson et al., 2018*). Both responses share a significant number of overlapping genes, and are mediated by transcription factors in the GATA family; in line with the involvement of these regulators in amino acid signalling and lifespan modulation across eukaryotes (*Dobson et al., 2018*).

While these data are starting to paint an increasingly detailed picture of nutrient-dependent regulation in females, the lack of information on males severely limits our understanding of how diet shapes metabolism and life history decisions. For example, it is not clear to which degree the regulation identified in females reflects their specific dietary requirements and physiology. Further, we cannot tell whether males and females differ in their general metabolism and its nutrient-dependent regulation, or whether diet responses are largely shared, and sex-specific effects limited to the regulation of reproductive investment. Interestingly, perturbing the IIS/TOR network in virgin flies has been shown to elicit sex-specific expression changes in males and females (*Graze et al., 2018*), but the link to nutrition and the effect on reproductive investment remains unclear. Addressing these questions is important because they have implications for the degree to which male and female physiology and its regulation are uncoupled and able to independently evolve. Thus, a shared physiology and diet-dependent regulation of metabolism across the two sexes would constrain the degree to which each sex is able to independently optimise its life-history decisions in response to the current nutritional environment.

Here, we are starting to address these fundamental questions by investigating male and female diet responses in gene expression. We study this in the context of shifts of nutritional composition (amino acid-to-carbohydrate ratio) between the male and female optima. This manipulation is more subtle than classic dietary restriction, given we are changing the quality of the diet whilst keeping caloric intake the same. This approach allows us to contrast, for each sex, an optimal and a non-optimal condition, as well as, across sexes, a more amino acid- and a more carbohydrate-rich diet. Furthermore, we can compare the female responses to a smaller, more quantitative shift in diet composition to existing data on responses to DR. We use nutritional geometry techniques to establish the male and female optimal diets in an outbred *D. melanogaster* population and then examine the transcriptomic responses of both sexes to the male-optimal diet (protein-to-carbohydrate ratio 1:4) and the female-optimal diet (2:1). We then assess the degree to which expression changes from male- to female-optimal diets are shared or divergent between the sexes, and how this relates to the function and regulation of genes.

Our analysis reveals that most of the core metabolic gene network is shared between the sexes, responding to diet changes in a sexually concordant manner. However, we also find smaller sets of genes where male and female responses diverge, either by being restricted to one sex or by males and females showing opposing diet-induced expression changes and observe that sex-limited reproductive genes are generally up-regulated on each sex's optimal diet. These results indicate that while males and females share a common, and concordantly regulated metabolic machinery, the sexes diverge in how nutritional information is translated into reproductive regulation. Further results allow us to link this divergent regulation to the Tor pathway. First, we find that our genes with diet-dependent regulation overlap with genes previously associated with responses to DR, rapamycin treatment and perturbation of the IIS/TOR network and known targets of the TOR pathway. Second, we can show experimentally that inhibiting TORC1 with rapamycin has a disproportionately negative effect on reproductive fitness on each sex's optimal diet. These results are compatible with the shared nutrient-sensing signal being inverted in males and females to produce diametrically opposed Tor-dependent regulation of reproduction in the two sexes.

## Results

### Dietary requirements and choice

We first examined the effects of diet composition on male and female fitness. We recovered previous results, finding that males and females differ significantly in their dietary requirements to maximise fitness (parametric bootstrap analysis: PB-stat = 78.002, p<0.001). For females, the number of eggs produced differed significantly between diets (Analysis of Variance, $F_7$ = 41.4703, p<0.001) and was maximised on the 2:1 (P:C) nutritional rail (*Figure 1* and *Figure 1—figure supplement 3*).

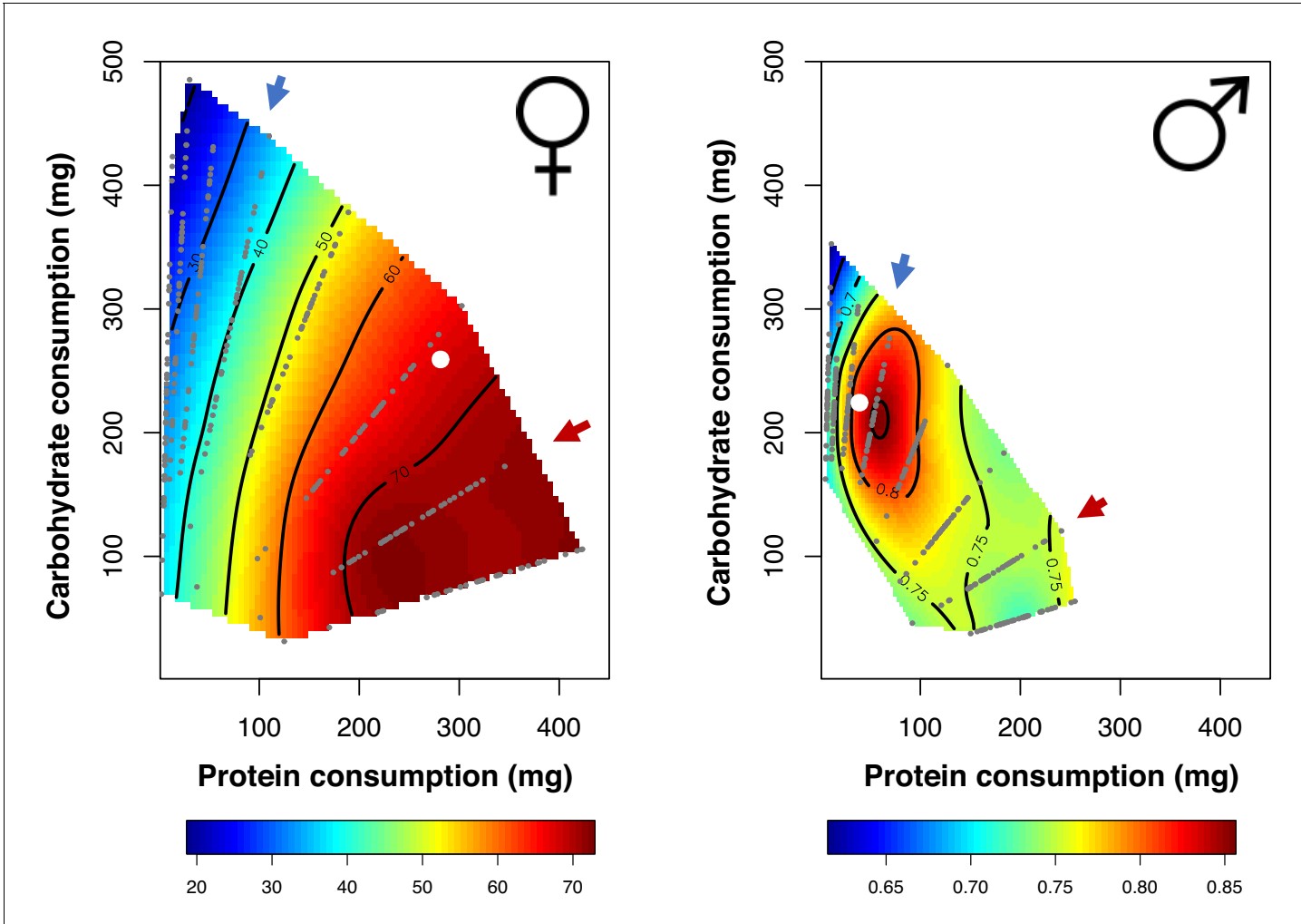

**Figure 1.** Nutritional landscapes for female (left) and male (right) fitness in the $LH_M$ population. Small grey dots represent the dietary coordinates of individual fitness measures. Dietary choices for each sex are also plotted (white dot). The red arrow denotes the female optimal nutritional rail (P:C = 2:1), whereas the blue arrow is the male optimal nutritional rail (P:C = 1:4). For each nutritional rail we samples 120 flies of each sex.
DOI: https://doi.org/10.7554/eLife.47262.003

The following figure supplements are available for figure 1:

**Figure supplement 1.** Experimental design for transcriptomic experiment.
DOI: https://doi.org/10.7554/eLife.47262.004

**Figure supplement 2.** Total liquid diet consumption.
DOI: https://doi.org/10.7554/eLife.47262.005

**Figure supplement 3.** Female fecundity (number of eggs laid) across dietary treatments.
DOI: https://doi.org/10.7554/eLife.47262.006

**Figure supplement 4.** Male competitive fertility across dietary treatments.
DOI: https://doi.org/10.7554/eLife.47262.007

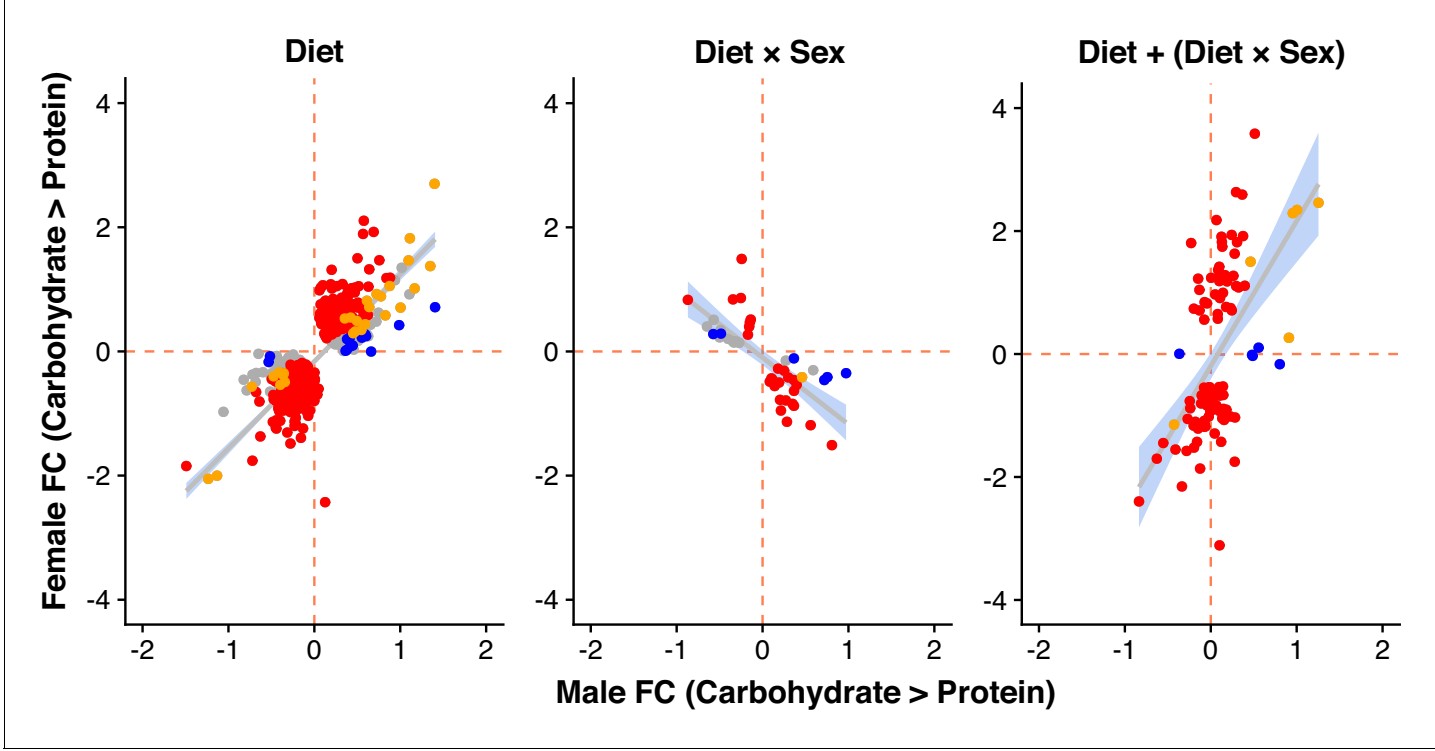

**Figure 2.** Male and female expression responses (log$_2$-fold change) for genes classified as showing only a diet effect (Diet), only a diet-by-sex interaction (Diet × Sex) or both (Diet + Diet × Sex). Expression changes are measured from the carbohydrate- to protein-rich diet. Colours represent genes with significant differential expression (at 5% FDR) only in females (red), only in males (blue), in both sexes (yellow) or in neither sex (grey).
DOI: https://doi.org/10.7554/eLife.47262.008

The following figure supplements are available for figure 2:

**Figure supplement 1.** Male and female expression responses (log$_2$-fold change) for genes classified as showing only a diet effect (Diet), only a InR-by-sex interaction (InR ×Sex) or both (InR + InR ×Sex) in the re-analysis of the *Graze et al. (2018)* dataset.
DOI: https://doi.org/10.7554/eLife.47262.009

**Figure supplement 2.** Female (left) and male (right) expression responses (log$_2$-fold change) in response to IIS/TOR perturbation (*Graze et al., 2018* dataset) and diet manipulation.
DOI: https://doi.org/10.7554/eLife.47262.010

Male competitive fertilisation success also differed between diets ($F_7$ = 3.5927, p<0.001), but peaked at the 1:4 ratio (*Figure 1* and *Figure 1—figure supplement 4*). Dietary choices also differed between the sexes ($F_2$ = 27.826, p<0.001). The choices of both sexes closely matched their previously established optimal composition, with females choosing to consume a more protein-rich diet than males (*Figure 1*). We also found that females, on average, tend to consume more liquid food than males but this relationship depends on the diet (sex ×diet: $F_7$ = 5.66, p<0.001, *Figure 1—figure supplement 2*).

## Transcriptional responses to diet

We measured gene expression in males and females maintained on food of either the female-optimal (2:1) or male-optimal (1:4) protein-to-carbohydrate ratio. We separately analysed transcriptomic responses in genes that were expressed in both males and females (hereafter 'shared genes', N = 8888) and those that showed sex-limited expression ($N_{male-limited}$ = 1879 and $N_{female-limited}$ = 165, see *Supplementary file 2* for full gene lists). For each shared gene, we tested for the effect of sex, diet and the sex-by-diet interaction on expression level. As expected, we found evidence for sex-differences in expression for a large number of genes (a total of 8318 genes with significant sex effect). In addition, we found large-scale transcriptomic responses to diet (806 genes with significant diet effect). Despite the large differences between male and female dietary requirements and food choices, the largest part of the transcriptional responses to diet is shared between the sexes

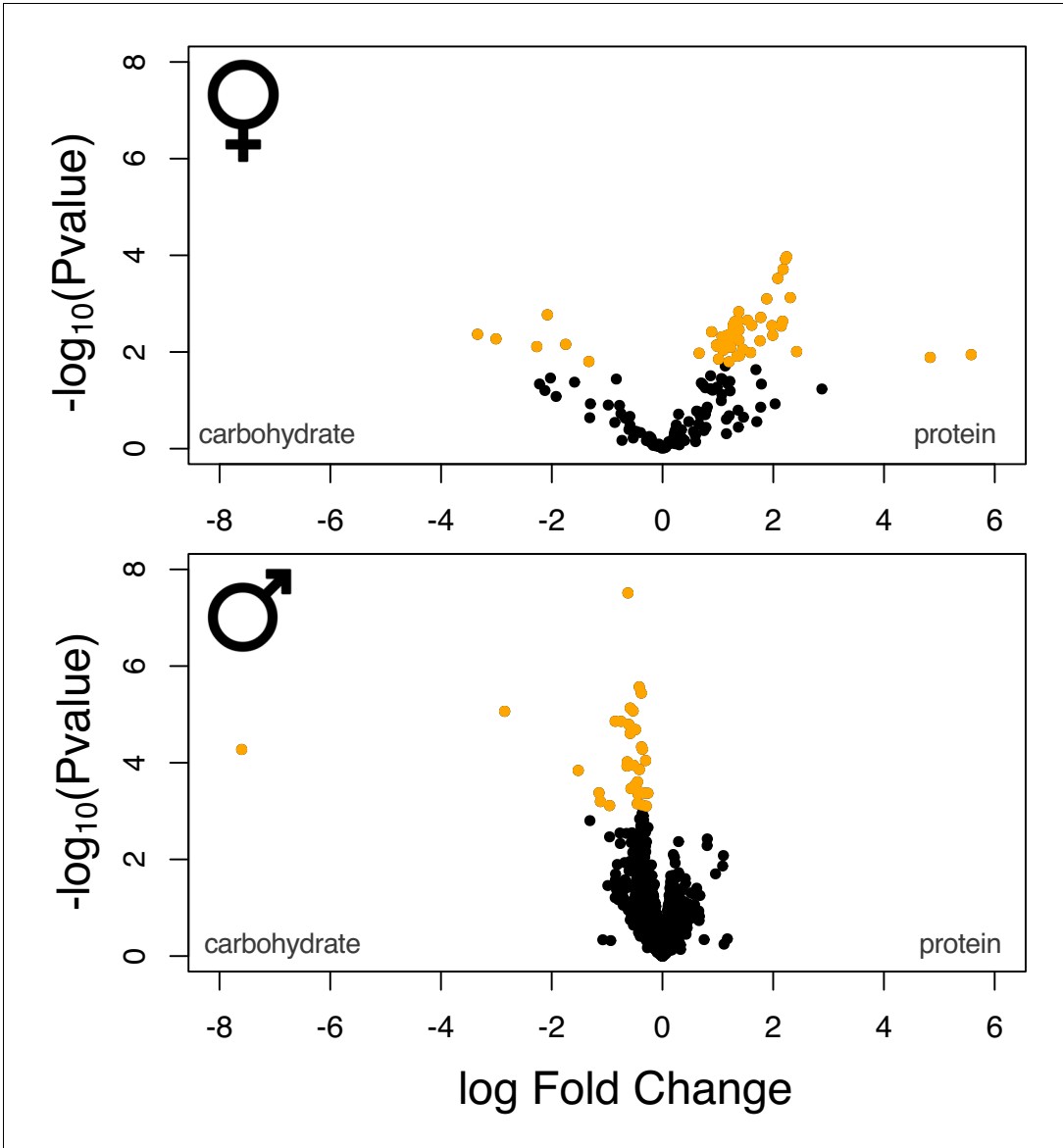

**Figure 3.** Volcano plot of the sex-specific gene sets. Yellow data points denote genes that were identified as differentially expressed at a 5% FDR cut-off.

DOI: https://doi.org/10.7554/eLife.47262.013

(significant diet effect but no interaction, category 'D' in *Table 1*, 639 genes). Here, males and females show parallel shifts in expression (although in most cases from a sexually dimorphic baseline expression) when reared on high-carbohydrate vs. high-protein food, and fold-changes between the two diets are strongly positively correlated between males and females (*Figure 2*; r = 0.76, p<0.001).

In addition to these sexually concordant responses, however, we also find a significant number of genes where the sexes show different responses to diet shifts (significant sex-by-diet interaction). For some of these genes, male and female expression change in opposing directions (category 'D × S' in *Table 1*, 51 genes). Thus, genes that are more highly expressed on a protein-rich diet in one sex are more lowly expressed on that diet in the other sex, resulting in negatively correlated fold-changes in the two sexes (*Figure 2*; r = −0.75, p<0.001). For another, larger group of genes (category 'D+D × S', 116 genes), both sexes tend to show expression shifts in the same direction (significant diet effect) but differ in the magnitude of their responses (significant interaction term).

**Table 1.** Shared transcriptomic response.

Number of genes that are influenced by sex (S), diet (D), and their interaction (D × S). From this method, we were able to cluster genes into three main categories. Categories highlighted in orange encompass genes that show an additive effect to diet (D), whereas clusters in blue show interactive effects (D × S). Green rows show both additive and interactive effects (D+D × S).

Significance (FDR < 0.05)

| S | D | D × S | n. genes |
|---|---|---|---|
| - | - | - | 545 |
| - | - | Y | 3 |
| - | Y | - | 18 |
| - | Y | Y | 4 |
| Y | - | - | 7537 |
| Y | - | Y | 48 |
| Y | Y | - | 621 |
| Y | Y | Y | 112 |

DOI: https://doi.org/10.7554/eLife.47262.011

These genes typically show a large expression response in one sex, but only a small or no response in the other sex, with overall a lower correlation of fold changes across sexes (r = 0.53, p<0.001). For the most part, the dominant expression change occurs in females, but there is a small number of genes where only male expression responds to diet (*Figure 2*).

We next analysed diet responses in genes with sex-limited expression. Similar to shared genes, we also observed significant expression changes in response to diet (*Table 2*). Thus, 56 out of 165 female-limited genes showed significant expression change between carbohydrate- and protein-rich diets. The majority of these (50 genes) had higher expression on the protein-rich diet preferred by females, while only a small number (six genes) had higher expression on the less beneficial carbohydrate-rich diet (*Figure 3*). In males, 30 out of the 1879 genes with male-limited expression showed significant diet responses. All of these had higher expression in the males' preferred carbohydrate-rich diet, compared to the less beneficial protein-rich media (*Figure 3*). Taken together, these results show that both sexes respond to their nutritional environment by upregulating sex-limited genes on their respective optimal diets.

## Functional enrichment of dietary responses

We used several approaches to investigate the functions of the genes showing diet responses. First, we performed Gene Ontology (GO) enrichment analyses for the shared genes of the three categories (D, D × S, D+D × S) defined above. We found distinct and significant enrichment in each class, with a predominance of GO terms relating to neuronal and metabolic biological processes (*Figure 4*). Second, we took a more targeted approach and analysed male and female expression changes for genes with specific GO annotations. With this analysis we aimed to assess how metabolic genes responded to diet manipulation, compared to the rest of the genome. For this, we fist created a 'baseline' of gene expression by extracting a list of genes that fall under the parent term 'Biological Process' (GO:0008150). From that list, we then removed the genes in the offspring category 'Metabolic Process' to create a set of genes performing biological functions unrelated to metabolism. We then compared this baseline to genes that fell within the following GO categories: 'Metabolic Process' (GO: 0008152), 'Glycolysis' (GO:0006096) and 'TCA cycle' (GO:0006099). The latter two were chosen as core processes in carbohydrate and protein metabolism. For the sets of genes in each of these categories that showed shared expression across the sexes, we found positive correlations between male and female fold changes between the two diet treatments ($R_{MP}$ = 0.35, $R_{GLY}$ = 0.74, $R_{TCA}$ = 0.6, *Figure 5A*). These correlations were significantly more positive than the (also slightly positive) correlation observed in the non-metabolic baseline gene set, despite the fact that correlations for the small Glycolysis and TCA gene sets have wide confidence intervals (*Figure 5B*). This indicates that, even though there is a general shared response to diet between males and females,

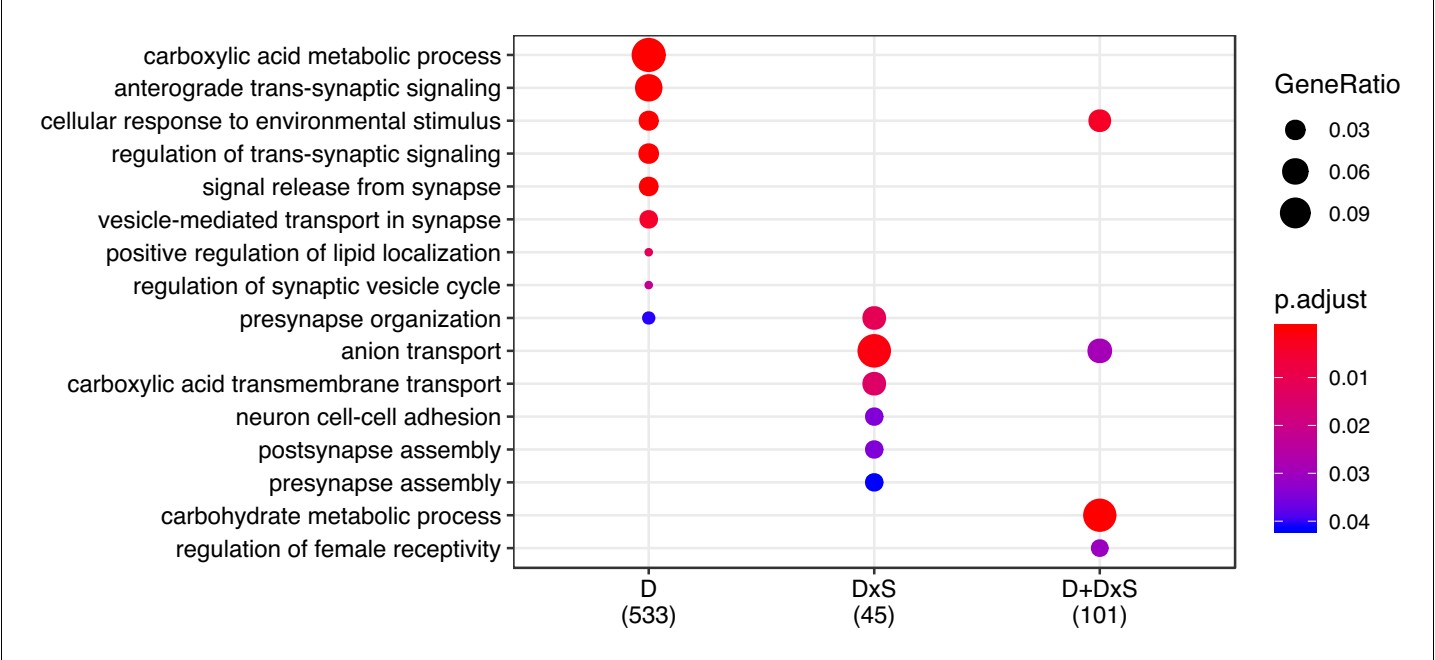

**Figure 4.** GO enrichment for the shared transcriptomic response. Enrichment for 'biological process' was performed for all categories, and p-values were adjusted for FDR < 0.05 ('p.adjust').

DOI: https://doi.org/10.7554/eLife.47262.014

male and female responses are more similar in genes involved in core metabolic processes than the rest of the genome.

For the sex-limited differentially expressed genes, we unsurprisingly found an enrichment of GO terms involved in reproduction (*Figure 6*). In females, differentially expressed genes were enriched for functions associated with egg production (chorion-containing eggshell formation), but also hormonal control (ecdysone biosynthetic pathway and hormone synthetic pathway). Male differentially expressed genes were enriched for sperm function (sperm competition). Since responses in both sexes consisted predominantly of up-regulation of genes under their respective optimal diets, these results show that for both males and females, the expression of reproductive genes is increased in the condition that maximises the fitness of that sex.

## Regulation of dietary responses

In order to infer the regulators that drive the observed expression responses to diet, we searched for enrichment of transcription factor binding motifs upstream of the genes in the three categories. Our analyses revealed significant enrichment of regulatory motifs in each group (see *Supplementary file 3* for a full list). Genes that showed only significant diet responses (concordant response between the sexes, D), presented an overrepresentation of binding motifs for the transcription factors *CrebB* and *lola*. Genes that showed opposing changes in males and female (D × S) were enriched for motifs for *vri* and GATA transcription factors (*grn*, *pnr*, *srp*, *GATAd*, *GATAe*).

**Table 2.** Sex-specific transcriptomic response.
Number of genes that are differentially expressed when moving from a carbohydrate-rich environment to a protein-rich environment in females and males (FDR < 0.1).

| Sex | Contrast | UP | Ns | DOWN | Total |
|---|---|---|---|---|---|
| Female | Carb → Protein | 50 | 109 | 6 | 56 |
| Male | Carb → Protein | 0 | 1845 | 34 | 34 |

DOI: https://doi.org/10.7554/eLife.47262.012

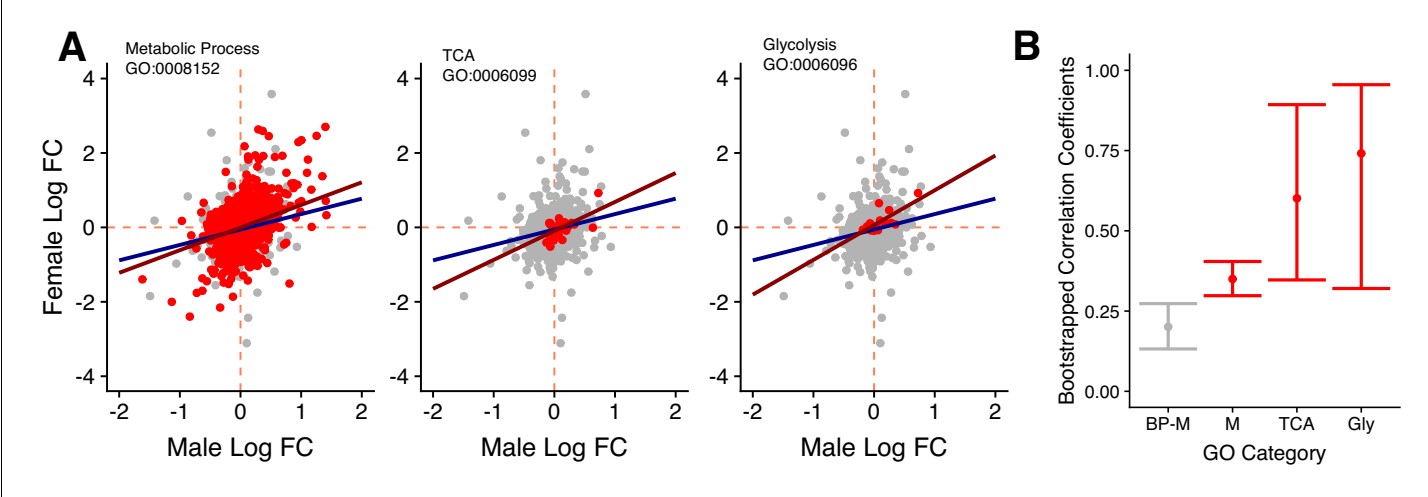

**Figure 5.** Gene expression for specific GO categories. (**A**) Male and female log₂ fold changes in gene expression going from carbohydrate to protein diets for selected GO terms. Red data points are genes that are found within the chosen GO terms (Metabolic process, TCA cycle and Glycolysis), whereas grey datapoints all other genes annotated with the first-level parent term 'biological process'. This background set provides a transcriptome-wide expression baseline between the sexes. Lines represent the regression of female on male log₂ fold change for the target term (red) and the background set (blue). (**B**) Bootstrapped correlation coefficients selected GO categories (red) and the baseline (grey).
DOI: https://doi.org/10.7554/eLife.47262.015

Finally, genes that showed diet responses largely restricted to one sex (D+D × S) were enriched primarily for GATA motifs, irrespectively of whether the response occurred predominantly in females or predominantly in males. Female-specific genes were mostly enriched for the transcription factors *Blimp-1*, *slbo* and *Dfd*, whereas male-specific genes were enriched for regulation by *pan* and *Sox*.

## Overlap with previously described diet and nutrient-signalling responses

We used comparisons to previously published transcriptomic datasets to relate the responses to shifts in diet quality observed here to those triggered by dietary restriction and perturbations of nutrient signalling. First, we compared genes in our three categories of diet-dependent regulation overlapped significantly with sets of genes that change expression in response to dietary restriction and rapamycin in females, analysed separately for brain, thorax, gut, and fat body (*Dobson et al., 2018*). We found significant overlap in the majority of comparisons made (*Table 3A and B*). Non-significant results were only obtained for some comparisons involving the list of genes in the D × S category, where males and females show opposing responses to diet. While this might reflect biological reality, it has to be noted that the numbers of genes—and hence statistical power to detect overlap—are smallest in the D × S category. Overall, the results of these comparisons demonstrate that transcriptional responses to the more subtle changes in dietary composition that we apply here generally mirror those that have previously been observed under dietary restriction.

We then compared our gene categories with a dataset from heads of virgin males and females in which *IIS/TOR* signalling had been perturbed by expressing a dominant-negative allele of the insulin receptor InR[DN] (*Graze et al., 2018*). Reanalysing this dataset (see Materials and methods) we obtained a list of genes that were altered by an IIS/TOR perturbation across both sexes (N = 5200 genes) similar to the results obtained in the original paper. However, subjecting the data to an analysis analogous to that we performed on our own, we further found that IIS/TOR perturbation causes expression changes similar to those observed for our diet treatments. Thus, a large number of genes show concordant responses to altered insulin signalling in males (significant InR effect) and females, while a second set shows opposing responses (InR-by-sex interaction, InR×S) and a third shows largely sex-specific responses (InR+InR×S) (*Figure 2—figure supplement 1*, *Supplementary file 4*). Furthermore, we detect parallelism in the effects of diet manipulation and InR perturbation on several levels. At the most basic level, the genes that are significantly affected by IIS/TOR perturbation

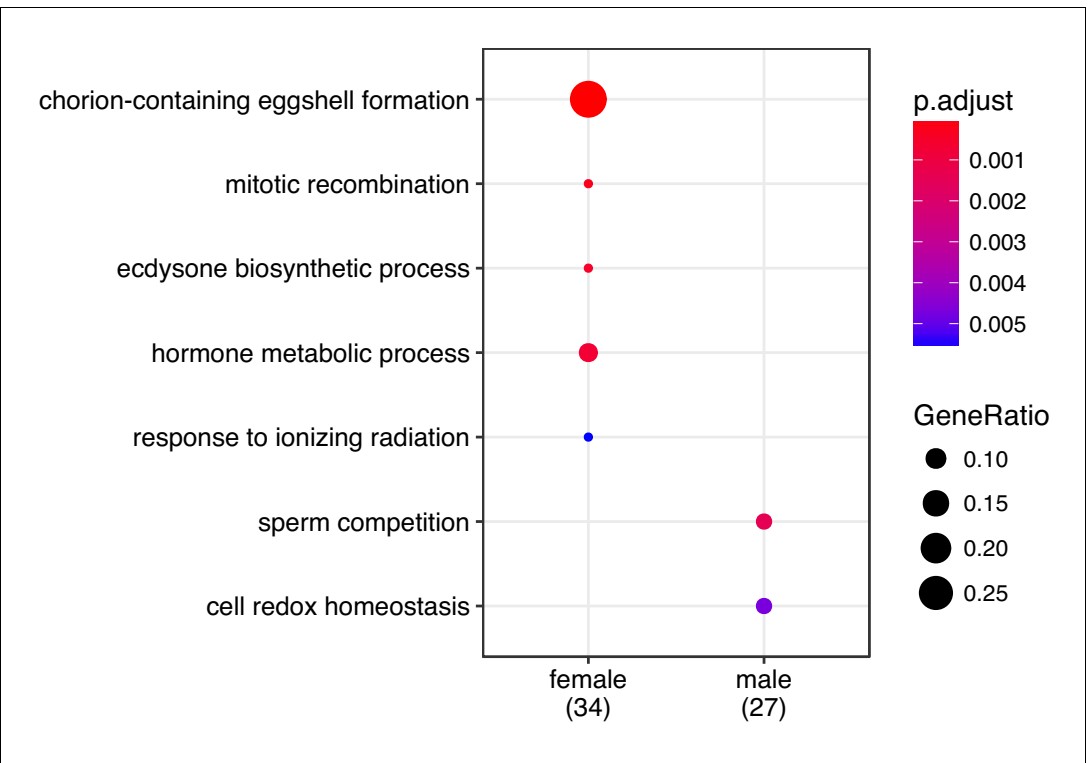

**Figure 6.** GO enrichment for sex-specific genes. Enrichment for differentially expressed genes was performed using 'biological process' and p-values were adjusted for FDR ('p.adjust').
DOI: https://doi.org/10.7554/eLife.47262.016

overlap significantly with the genes that are significantly affected by diet quality (489 genes observed, 351 expected, 39% excess, Fisher's exact test, p < 0.001). Second, genes that show a significant diet effect ('D') are more likely to also show a significant effect of InR perturbation ('InR') (436 genes with both terms significant, 37% excess, Fisher's exact test, p < 0.001) and genes with a significant diet-by-sex interaction are more likely to also show a significant InR-by-sex interaction (51 genes, 108% excess, Fisher's exact test, p < 0.001). Third, a full comparison based on a contingency table containing all possible combinations of classes also showed a significant correspondence (Chi-squared test, $X_9^2$ = 248.53, p < 0.001), with excess overlap in most combinations of classes as well in genes that are classified in neither analyses (*Supplementary file 1* — Table 5). And finally, fold changes in male and female gene expression in response to IIS/TOR perturbation correlate positively with those in response to diet manipulations (see *Figure 2—figure supplement 2*, *Supplementary file 4*), despite the fact that the two datasets analyse different tissues (head vs. whole body). These results indicate that manipulating diet quality and manipulating IIS/TOR signalling produces parallel and overlapping expression responses.

We also investigated the overlap between our diet-responsive genes and genes that have been identified as TORC1-regulated due to their dependence on REPTOR and REPTOR-BP (*Tiebe et al., 2015*). While based on expression in S2 cells only, this to our knowledge is the best characterised set of TOR-responsive genes. In line with the similarity between expression responses to diet and IIS/TOR-manipulation described above, we find significant overlap between our gene categories and genes with REPTOR- or REPTOR-BP–dependent expression, specifically in our category that responds to diet ('D', 28 genes) and our sex-biased category ('D+D × S', nine genes, *Table 3C*, *Supplementary file 4*).

## Effect of rapamycin treatment on diet-specific fitness

The overlap with previously described responses raises the potential for the IIS/TOR network, and specifically TORC1, mediating the diet-dependent phenotypes that we observe here. This appears

**Table 3.** Gene overlap between our three categories (D, D × S, D+D × S) and results from three previously published papers.

The first study (A+B) examines female transcriptomic response to dietary restriction and rapamycin across six different tissues (*Dobson et al., 2018*). The second study (C) characterises genes that respond to TORC1 inhibition via the transcription factors REPTOR and REPTOR-BP (*Tiebe et al., 2015*). In italics we show the total number of genes in that category, with bold counts showing the significant (p<0.05) overlaps between two categories. Overlap is assessed with Fisher's exact tests, p-values are provided below the counts.

**A. Dietary Restriction**

|  | Brain (167) | Thorax (193) | Gut (25) | Fatbody (358) |
|---|---|---|---|---|
| D (639) | 27 p<0.001 | 51 p<0.001 | 14 p<0.001 | 58 p<0.001 |
| D × S (51) | 5 p=0.0026 | 5 p=0.0048 | 0 p=1 | 7 p=0.0041 |
| D+D × S (116) | 10 p<0.001 | 19 p<0.001 | 3 p=0.004 | 20 p<0.001 |

**B. Rapamycin**

|  | Brain (58) | Thorax (38) | Gut (76) | Fatbody (222) |
|---|---|---|---|---|
| D (639) | 14 p<0.001 | 9 p=0.0012 | 17 p<0.001 | 57 p<0.001 |
| D × S (51) | 5 p<0.001 | 2 p=0.02 | 2 p=0.07 | 3 p=0.13 |
| D+D × S (116) | 6 p<0.001 | 7 p<0.001 | 4 p=0.017 | 16 p<0.001 |

**C. TORC1**

|  | REPTOR/REPTOR-BP (212) |
|---|---|
| D (639) | 28 p=0.019 |
| D × S (51) | 1 p=0.78 |
| D+D × S (116) | 9 p=0.0068 |

DOI: https://doi.org/10.7554/eLife.47262.017

plausible for the modulation of female fecundity in response to diet, where the role of TORC1 is well established, but has not been assessed in males. We therefore directly tested the phenotypic effect of varying doses of rapamycin and its interaction with diet, on our proxies for male and female fitness. Our experiment showed that, across the two sexes, rapamycin leads to a reduction in reproductive output (rapamycin effect: p<0.001, *Figure 7* and *Figure 7—figure supplement 1*, *Supplementary file 1* - Table 4). More importantly, however, we also found a significant interaction between diet and rapamycin treatment that was shared across males and females, where rapamycin lead to a larger reduction in reproductive output on each sex's optimal diet (sex ×rapamycin: p=0.001). Finally, our experiment revealed possible quantitative differences between the sexes in the effect of rapamycin on reproduction (sex ×rapamycin × diet: p=0.068); while the effect of the treatment in females correlated roughly with the dose administered, males showed a threshold response where all rapamycin levels in the optimal diet resulted in a reduction in reproductive output to the level observed on the non-optimal diet.

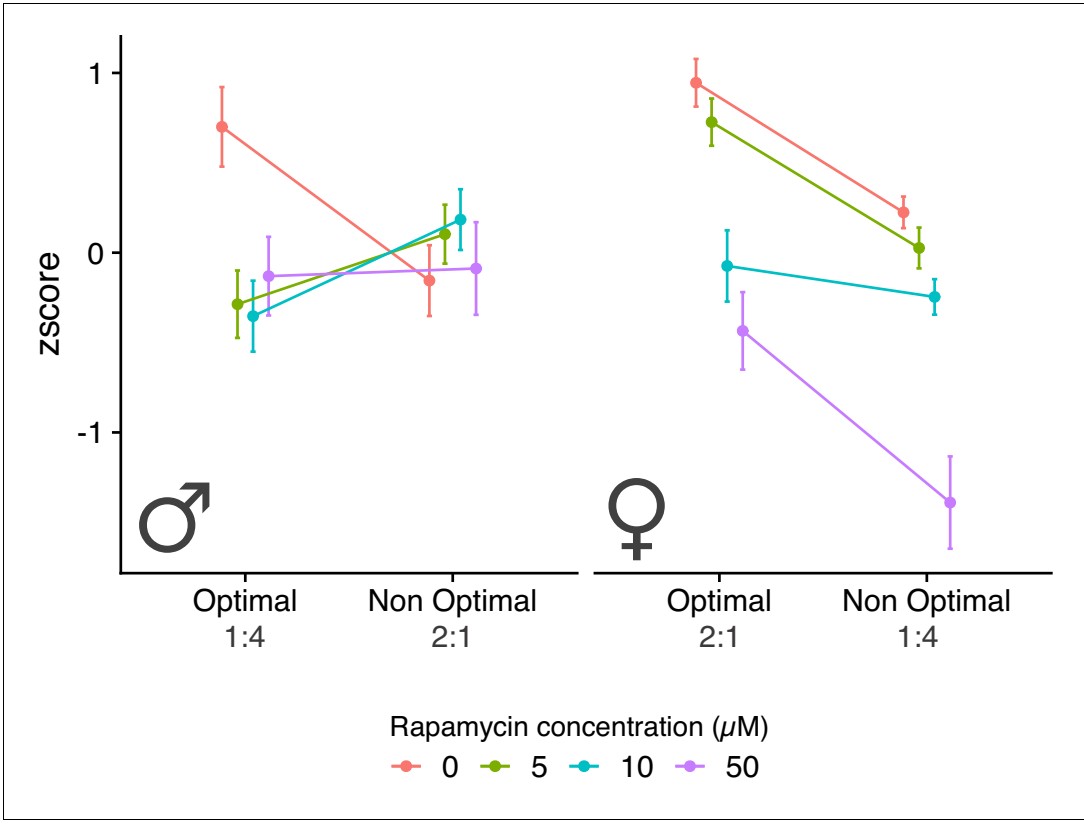

**Figure 7.** Male and female fitness measures across the two diets and for rapamycin treatments. Sample size for each experimental treatment is 60 flies.
DOI: https://doi.org/10.7554/eLife.47262.018
The following figure supplement is available for figure 7:

**Figure supplement 1.** Sex-specific fitness measured across both diets and four rapamycin treatments.
DOI: https://doi.org/10.7554/eLife.47262.019

## Discussion

Our study examined the transcriptomic response of male and female *D. melanogaster* to variation in dietary composition, being exposed to either a male-optimal (protein-to-carbohydrate ratio 1:4) or a female-optimal (2:1) diet. Our results provide interesting insights into nutritional effects on male and female fitness in relation to sex- and diet-dependent expression levels, function and regulation. We show that both sexes share a large metabolic core transcriptome that is regulated in a sexually con-cordant way. Nevertheless, smaller parts of the transcriptome are sex-specifically regulated to diet, including sex-limited reproductive genes. Together with the observed effects of rapamycin in the two sexes, this suggests that male and female reproduction is inversely regulated in response to diet composition.

### A shared metabolic core transcriptome

Our analyses demonstrated the existence of a core metabolic transcriptome that shows sexually con-cordant regulation in response to diet. Overall, expression fold changes from carbohydrate- to pro-tein-rich food among metabolic genes are positively correlated between the sexes, and significantly more so than for the transcriptomic background. This indicates that gene expression in males and females responds generally similarly to changes in dietary composition. In line with this interpreta-tion, the large majority of genes with diet-dependent expression show significant changes only in response to diet, independently of sex (639 out of 806 genes, 79%). Functionally, genes in this core metabolic transcriptome are enriched for carboxylic acid metabolism and neurological biological processes. Carboxylic acid metabolism is an integral part of both protein and carbohydrate

processing—for instance, part of the components of amino acids are carboxylic acid sidechains. The prominence of neurological biological processes, on the other hand, supports the notion of a neural gut-brain connection that is conserved evolutionarily (*Kaelberer and Bohórquez, 2018*) and shared between the sexes. Specifically, the sensory mechanisms in the gastrointestinal tract convey information about the nutritional status to regulate satiety (and thereby feeding behaviour), metabolism, and digestion (*Kaelberer et al., 2018*) in a way that is similar between males and females.

We were also able to infer key regulators of sexually concordant, diet-dependent gene expression, using motif enrichment tools. Upstream regions of genes with sexually concordant diet responses were enriched for motifs of two main transcription factors *CreB* and *lola* transcription factors. *CrebB* is involved in diurnal rhythms and memory formation (*Bittinger et al., 2004*; *Kogan et al., 1997*), but also in energy homeostasis and starvation resistance, mediated by insulin signalling (*Wang et al., 2008*). The *lola* transcription factor, on the other hand, is mainly involved in axon guidance in *Drosophila* (*Horiuchi et al., 2003*; *Goeke et al., 2003*). But interestingly, some protein isoforms have also been associated with octopamine synthesis pathways which are essential for nutrient sensing (*Dinges et al., 2017*).

## Sex-specific diet responses in gene regulation

Besides the large, shared core metabolic transcriptome, we also identified smaller groups of genes with sex-specific expression responses to diet. A first group showed opposing diet responses in males and female (D × S, 51 out of 806 genes, 6.3%). These genes are enriched for transport functions and synapse assembly/organisation. One of our candidate antagonistic genes is *fit* (*female-specific independent of transformer*). Known to be sexually dimorphic in expression, *fit* has been found to be rapidly upregulated in male heads during the process of male courtship and mating, along with another antagonistic candidate *Odorant binding protein 99b, Obp99B* (*Ellis and Carney, 2010*; *Carney, 2007*). Interestingly, *fit* has also been implicated in protein satiety in a sex-specific manner (*Sun et al., 2017*). Following the ingestion of protein-rich food, *fit* expression increases in both sexes (although more so in females than males), but only supresses protein appetite in females (*Sun et al., 2017*). Both *fit* and *Obp99B* were found to be significantly altered in a sex-specific way when flies were starved, further cementing their role in nutrient response (*Fujikawa et al., 2009*). Together with previous work, our results therefore cement the tight link between nutritional sensory mechanisms and reproduction, however this response is sex-specific.

Another group of genes showed mostly responses in one sex (D+D × S, 116 genes, 14.4%). Most of the genes observed in this category show expression changes in females (with little change in male expression levels) and are mainly involved in carbohydrate metabolism and female receptivity. One notable gene in this category is the transcription factor *doublesex,* which plays a key role in sexual differentiation and the regulation of sex-specific behavioural traits (*Shirangi et al., 2006*). Expression levels of this gene are higher in females that are fed a high-protein diet (unless the difference in *dsx* mRNA levels is due to growth in a sexually dimorphic, and hence *dsx*-expressing, tissue type). Of interest among the few genes with male-limited diet response (*Figure 2*) is *Adenosylhomosysteinase* (*Ahcy*), which we find males to express at lower levels on the carbohydrate-rich (optimal) diet. *Ahcy* is involved in methionine metabolism and has been linked to male lifespan regulation. *Ahcy* knock-outs were shorter lived, while knock-outs for two putative *Ahcy*-repressors extended male life- and health-span (*Parkhitko et al., 2016*). These effects are in line with the under-expression we observe on high carbohydrate, under the assumption that greater investment in current reproduction is associated with decreased lifespan (which may not generally hold in the context of nutrient manipulation; *Jensen et al., 2015*).

Both the genes with opposing (D × S) and those with sex-limited diet-dependent regulation (D +D × S) show significant enrichment for GATA transcription factors. This class of transcription factors has been previously implicated in female nutritional and reproductive control. For example, the ovary-specific *dGATAb* binds upstream of both yolk protein genes *Yp1* and *Yp2* (*Lossky and Wensink, 1995*). GATA-related motifs have also previously been shown to be enriched in genes showing differential expression in response to DR and rapamycin treatment in female flies (*Dobson et al., 2018*). The shared regulation is further supported by the fact that the diet-responsive genes we identify here also overlap significantly with those previously inferred to respond to DR- and rapamycin-treatment. These results suggest that changing the *quality* of the diet elicits a similar response as changing the *quantity* via protein dilution. This may not be surprising, if DR is considered a response

mainly to the quantity of protein ingested (*Lee et al., 2008*; *Grandison et al., 2009*), and fits with previous work that found the ratio of macronutrients—not caloric intake—to be the main determinant of healthy ageing in mice (*Solon-Biet et al., 2014*). However, the overlap highlights that DR-phenotypes are not an all-or-nothing response but instead are part of a continuum of life history adjustments in response to how suitable the dietary environment is for current reproduction.

## Diet-specific regulation of male and female reproduction

We also found diet responses in reproductive genes that are exclusively expressed in either males or females. Regulation largely reflects diet-dependent reproductive investment, with most genes being more highly expressed on a sex's optimal diet with lower expression on the suboptimal diet. In females, a significant number of these genes are involved in egg production and thus linked to diet-dependent reproductive investment (*Trivers and Campbell, 1972*). Also among the genes is *insulin-like peptide-7* (*dILP-7*), one of a family of peptides known to having the functional as hormones and neuropeptides (*Sisodia and Singh, 2012*) involved in nutrient foraging control (*Shim et al., 2013*). More specifically, *dILP-7* is expressed in neurons that play an active role in female fertility. These neurons have been linked with the egg-laying decision process (*Yang et al., 2008*; *Lihoreau et al., 2016*) and *dILP-7* is among a number of genes show sexually dimorphic expression in these neuronal cells (*Castellanos et al., 2013*). Interestingly, IIS/TOR perturbation also results in sex-specific changes in dILP peptides (dILP2, 3, 5 and 6) in the head (*Graze et al., 2018*) (where dILP7 is not expressed; *Nässel and Broeck, 2016*).

Mirroring expression responses in females, we also find higher expression of reproductive genes on the optimal diet in males. This is surprising—based on the view that male fitness is limited by the acquisition of mates and the supposedly low investment required for sperm production (*Trivers and Campbell, 1972*), one could expect that males do not modulate their reproductive investment in response to the nutritional environment but remain primed to maximally exploit any mating opportunity. Assuming that expression of these genes reflects reproductive investment, the fact that they do respond to the nutritional environment suggests that male reproductive strategies are maybe more subtle, and their investment more costly, than previously appreciated. This is plausible, as work on other insects has shown that the production of high quality sperm is costly (*Bunning et al., 2015*) (but courtship activity does not appear to carry a significant cost, at least in fruit flies; (*Flintham et al., 2018*).

Superficially, it may seem obvious that male and female reproductive genes are upregulated on each sex's respective optimal diet. In the presence of a largely shared and concordantly regulated metabolic machinery, however, this pattern implies that the output of nutrient sensing pathways is used in different, and potentially inversed ways in males and females. While our analyses do not allow us to identify the exact point of reversal within the regulatory hierarchy, our data provide some interesting insights. First, it is noteworthy that GATA transcription factors are inferred to be regulating genes that show a wide range of expression patterns, being enriched among genes with opposing expression changes in males and females (the D × S set), as well as those that show largely sex-limited responses (D+D × S). This could imply that the main role played by these factors is to convey information about the metabolic and nutritional state of the animal (similar to homeotic genes in development), which is then incorporated combinatorially with additional factors to give rise to the sex- and diet-specific expression patterns that we observe.

Second, several lines of bioinformatic evidence suggest that the expression changes that we describe here are at least in part regulated by IIS/TOR signalling. Thus, the genes that we find to respond to diet manipulation significantly overlap with genes affected by manipulation of IIS/TOR signalling as described by *Graze et al. (2018)*, a dataset that our reanalysis reveals to show a similar structure of genes with sexually concordant, sexually opposing and sex-biased expression changes. This pattern and the overlap with our data is all the more noteworthy as Graze et al. assessed the effect of IIS/TOR perturbation in virgin flies, where males and females have more similar dietary requirements, and hence presumably more similar physiological states, than in mated flies (*Camus et al., 2018*). In addition to showing parallels with IIS/TOR-dependent expression, our diet-dependent genes also significantly overlap with the arguably best-defined set of TORC1-dependent genes currently available (*Tiebe et al., 2015*). These results suggest that diet-dependent expression responses, and their sex-specific differences, are mediated by IIS and the TOR pathway.

This conclusion is corroborated by the results of our experiment combining diet manipulation with rapamycin treatments, which are consistent with TORC1-dependent upregulation of reproduction on optimal diets in both sexes. Here we find that while rapamycin generally lowers reproductive output, this effect is more pronounced on the respective optimal diet of each sex. This is expected in females, where a large body of work implicates the IIS/TOR network in life-history shifts between reproduction and longevity (*Wullschleger et al., 2006*). Accordingly, a nutritionally favourable environment should lead to increased TORC1 activity and elevated reproductive output. What our data show, however, is that a parallel effect of increased reproduction on the optimal diet is detectable in males, even though the composition of that diet is the one that is unfavourable in females, leading to low TORC1 and reduced reproduction. Across the sexes, TORC1 activity would thus not reflect a specific dietary composition but a measure of nutritional optimality and regulate reproductive investment accordingly.

We note that, while tantalising, these inferences will require further careful validation. Due to the focus on females, diet-dependent regulation of male reproduction has been little explored. Knockdown of *Tor* and *raptor* in males has been found to result in an accumulation of germline stem cells, combined with deficient differentiation (*Liu et al., 2016*). Future work will need to assess the effect of these changes on male reproductive output and, more importantly, whether and how the signal of the nutrient sensing mechanisms that feed into the Tor pathway are modulated in a sex-specific way. Independently of how the regulatory reversal is achieved mechanistically, our data also suggest that the relationship between the composition of the diet consumed and reproductive output does not merely reflect the passive effect of metabolic conversion rates from nutritional components to gametes and energy but is at least in part the result if an active regulation of immediate reproductive investment. This has important implications for our interpretation of variation in diet-specific reproductive success, which has been documented in the population studied here (*Camus et al., 2017*). Thus, variation between genotypes in the dietary composition that maximises, for example, male reproductive fitness is therefore probably at least partly caused by genetic variation in how nutrients are sensed or how this sensory output is used to regulate reproductive investment. Studying this variation in more detail will provide a fruitful avenue to better understand the regulatory mechanisms involved, as well as the selective forces that shape variation in its components.

## Materials and methods

### Fly stocks and maintenance

We used the *D. melanogaster* laboratory population LH$_M$ for our experiments. This has been sustained as a large outbred population for over 400 non-overlapping generations (*Chippindale et al., 2001*; *Rice, 1996*), maintained on a strict 14 day regime, with constant densities at larval (~175 larvae per vial) and adult (56 vials of 16 male and 16 females) stages. All LH$_M$ flies were reared at 25°C, under a 12 hr:12 hr light:dark regime, on cornmeal-molasses-yeast-agar food medium.

### Synthetic diet

We used a modified liquid version of the synthetic diet described in *Piper et al. (2014)*, that is prepared entirely from purified components to enable precise control over nutritional value (see *Supplementary file 1* Tables 1-3). Previous studies have used diets based on natural components, typically sugar as the carbon source and live or killed yeast as the protein source (*Piper and Partridge, 2007*). Such diets offer only approximate control over their composition, because the yeast-based protein component also contains carbohydrates and is required to provide other essential elements (vitamins, minerals, cholesterol, etc.) that vary in relative abundance. As a consequence, phenotypic responses to such diets cannot be straightforwardly interpreted in a carbohydrate-to-protein framework as they are confounded by responses to other dietary components. Our use of a holidic diet completely eliminates these problems without causing any apparent stress in the flies (*Piper et al., 2014*).

Eight isocaloric artificial liquid diets were made that varied in the ratio of protein (P, incorporated as individual amino acids) and carbohydrate (C, supplied as sucrose), while all other nutritional components were provided in fixed concentrations. Nutrient ratios used were [P:C] – 4:1, 2:1, 1:1, 1:2, 1:4, 1:8, 1:16 and 1:32, with the final concentration of each diet (sum of sugar and amino acids)

being 32.5 g/L. These ratios span the P:C ratio of the molasses medium on which the LH$_M$ population is maintained. Based on the media recipe used in our laboratory and the approximate protein and carbohydrate content of the ingredients, we estimate our standard food to have a P:C ratio of about 1:8. The diets in our experiments on the edges of our nutritional space, with the highest carbohydrate- or protein-bias, can thus considered to be 'extreme' in comparison to our standard laboratory media—even taking into account the fact that ratios in synthetic and organic diets may not be directly comparable, as nutrients in synthetic food appear to be more readily accessible (*Piper et al., 2014*).

For diet preference assay we used two diets; protein and carbohydrate. Each diet contained all nutritional components (vitamins, minerals, lipids) at equal concentration, with the protein diet containing amino acids and the carbohydrate diet containing sucrose. Preliminary experiments established that flies would not eat purified amino acids with the vitamin/mineral/lipid buffer, so we diluted our protein solution with 20% of a suspension of dried yeast extract, made at the same protein concentration as the synthetic solution (16.25 g/L). Given that yeast extract also contains sugars, the final protein diet then included 4% carbohydrate.

## Experiment 1a: Identification of male and female optimal diets

### Experimental setup and diet assay

Flies from each sex were collected as virgins using $CO_2$ anaesthesia. Three virgin females and three virgin males were randomly placed in individual vials containing culture medium (molasses-yeast-agar) with no added live yeast. Twenty vials of hextets were collected for each sex and diet treatment. Flies were left to mate for a period of 36 hr on molasses-yeast-agar medium. Following this period, they were split by sex (now fly triplets), and placed on 0.8% agar-water mixture. Agar-water vials provide water for the flies, but have no nutritional value. Flies were kept in these vials overnight before being supplied with a 10 µl (females) or 5 µl (males) microcapillary tube (ringcaps, Hirschmann) containing one of the eight allocated diets. These diets varied in their protein-to-carbohydrate ratios and captured the following nutritional rails (P:C): 4:1, 2:1, 1:1, 1:2, 1:4, 1:8, 1:16, 1:32. Capillary tubes were replaced daily, and food consumption for each fly trio was recorded for a total period of four days. Consequently, our experiment design consisted of 2 sexes and eight nutritional environments, with 20 vials of fly triplets comprising each experimental unit (two sexes × 8 diets×20 vials=320 vials, 960 flies). We chose to use capillary tubes of different sizes to maximise the accuracy of our diet consumption measurements and minimise evaporation errors. Larger capillary tubes increase evaporation rates; however, with a smaller capillary tube we ran the risk that flies would consume all of the food leading to a subsequent slight starvation response. For this reason, we found that a slightly larger capillary tube was ideal for females because they ate more than males in a 24 hr period. Using this approach, we found that flies never consumed all of the food from the capillary tubes. Flies were exposed to diet treatments in a controlled temperature room (25˚C), 12L:12D light cycle and high relative humidity >80%. The rate of evaporation for all diet treatments was measured by using five vials per diet that contained no flies, placed randomly in the constant temperature chamber. The average evaporation per day was used to correct diet consumption for evaporation. Following four days of feeding under these dietary regimes, flies were assayed for fitness.

### Male fitness assay

Male adult fitness was measured as the number of adult offspring produced in competitive mating trials. Previous work in our laboratory has shown this to be a robust measure of reproductive performance and, with lifetime adult production being largely determined by mating success in our population (*Pischedda and Rice, 2012*).

We used an experimental approach similar to *Collet et al. (2016)*, whereby focal experimental males competed with standard competitor males to mate with females. Following the experimental feeding period described above, a focal trio of males was placed into a new vial (provided with molasses-yeast-agar medium that did not contain live yeast, the main source of food for both males and females; *Sang, 1978*; *Colinet and Renault, 2014*), along with three virgin competitor males and six virgin females. The competitor males and the females were of LH$_M$ genetic background but homozygous for the recessive $bw^-$ eye-colour allele. Competitor flies were reared under the same

conditions as our experimental flies and were the same age as the experimental males. The flies interacted, and female flies produced eggs for a period of 24 hr, after which the adults were discarded from the vials. Eggs were left to develop for 12 days and the subsequent adult offspring in each vial were counted and scored and assigned paternity to either the focal experimental males (if the progeny had red eyes - wildtype) or the competitor males (if the progeny had brown eyes).

## Female fitness assay

Female adult fitness was measured as the number of eggs produced over a fixed period of time. This performance proxy is expected to correlate closely with other fitness measures, such as the total number of offspring (*Tanaka and Yamazaki, 1990*; *Hoffmann and Harshman, 1985*).

Following the feeding period, trios of mated females were placed in new agar vials and presented with three males from the LH$_M$ stock population. Flies were left to mate/oviposit for 18 hr in vials containing *ad libitum* food corresponding to their diet treatment provided via capillary tubes. All flies were removed after this 18 hr mating window. Following removal of the flies, the total number of eggs laid were determined by taking pictures of the agar surface and counting eggs using the software *QuantiFly* (*Waithe et al., 2015*).

## Statistical analyses

First, we sought to investigate the effects of diet on sex-specific fitness. Separate models were run for each sex, as the two datasets measured fitness in distinct ways. Female fitness was measured as total number of eggs produced within a 18 hr timeframe following a mating event. Given data followed a normal distribution, we used a linear model to analyse the data. Number of eggs was the response variable, with mating status, and diet plus their interaction as fixed factors. Male fitness was measured as the proportion of offspring sired from the focal male. For this we modelled the response as a binomial vector comprising the number of offspring sired by the focal male and the number sired by the competitor male and diet composition as a categorical fixed effect. To examine whether the sexes varied in the quantity they consumed of each diet, we used a linear model to investigate differences in dietary consumption. We modelled total food consumption as a response variable with sex, diet and their interaction as fixed effects. All models were performed using the *lm* function in R version 3.3.2 (*R Development Core Team, 2016*).

To examine nutritional fitness landscapes, we combined fitness values with diet consumption values for each sex. Before statistical analysis, we transformed the fitness data as male and female fitness were measured in different units. For this, we standardised them using Z-transformations for each sex across treatments. We used a multivariate response-surface approach (*Lande and Arnold, 1983*; *Chenoweth and Blows, 2005*) to estimate the linear and quadratic effects of protein and carbohydrate intake on male and female fitness. The linear gradients for protein and carbohydrate intake for each sex were estimated from a model containing only the linear terms. The nonlinear gradients for nutrient intake were obtained from a model that contained both linear and nonlinear terms. We used untransformed data to visualise nutritional landscapes, using non-parametric thin-plate splines implemented with the *Fields* package in R version 3.3.2 (*R Development Core Team, 2016*).

## Experiment 1b: Dietary preference assay

Alongside the dietary setup used for measuring diet-dependant fitness, we tested what flies preferred to eat, given the choice. For this, flies were supplied with two 5 μl microcapillary tubes (ringcaps, Hirschmann); one containing the protein solution and the other the carbohydrate solution. Capillary tubes were replaced daily, and food consumption for each fly trio was recorded for a period of three days. As a control, the rate of evaporation for all diet treatments was measured in six vials that contained the two solution-bearing capillary tubes but no flies and placed randomly in the controlled temperature room. Their average evaporation per day was used to correct diet consumption for evaporation.

## Statistical analysis

To determine if male and female dietary choices differed between the sexes, we used a multivariate analysis of variance (MANOVA). The main model had protein and carbohydrate consumption as

response variables, with sex as fixed effect. We performed subsequent univariate analysis of variance (ANOVA) to determine which nutrient(s) contributed to the overall multivariate effect. All analyses were performed using the *manova* function in R version 3.3.2 (*R Development Core Team, 2016*).

## Experiment 2: Transcriptional response

### Experimental setup

We followed the same experimental regime as previously stated, with the only exception of using two diets instead of eight (*Figure 1—figure supplement 1A*). In brief, flies were collected in hextets; three male and three female flies per vial, with 40 vials being setup. Following a period of 36 hr where flies had the opportunity to mate, they were split by sex and placed onto agar medium in triplets. Flies were allocated either a female-optimal diet (P:C = 2:1) or a male-optimal diet (P:C = 1:4). Liquid food was provided using a 10 ul capillary tube for females and a 5 ul capillary tube for males. Capillary tubes were replaced daily, and food consumption for each fly trio was recorded for a total period of four days. Following this period, flies were flash-frozen in their triplets.

We also set up 10 extra vials for each treatment alongside the RNA-Seq experiment where we re-measured male and female fitness and preference. This was to verify the repeatability of protocols for experiment 1 and 2.

### Sample collection and RNA extraction

We generated three biological replicates for each of the experimental treatments (females on female-optimal diet, females on male-optimal diet, males on female-optimal diet, males on male-optimal diet), a total of 12 samples. For each replicate sample, we pooled four triplets (a total of 12 flies) to ensure we collected sufficient amounts of RNA. Total RNA was extracted using the Qiagen RNeasy Minikits (Qiagen BV, Venlo, The Netherlands). This kit includes an on-column DNAse I digestion step. Quantity and quality of RNA was first inspected using a Nanodrop 2000 spectrophotometer (Wilmington, USA), and later verified using an Agilent Tapestation 2200 at the UCL Genomics facility.

### Sequencing and read mapping

Library construction and sequencing were handled by the UCL Institute of Child Health Genomics facility. cDNA libraries were constructed using the KAPA Hyper mRNA Library prep kit. cDNA from all 12 libraries was mixed at equal concentrations and these multiplexed samples were sequenced (43 bp paired-end reads) on four flowcell lanes on an Illumina Nextgen 500 instrument to an average of 18M reads per sample.

Having verified that there was no bias towards particular libraries across the sequencing lanes using the Illumina Basespace online server, we merged reads from all four lanes. Adaptors and low-quality base pairs were trimmed using trimmomatic v0.36 (*Bolger et al., 2014*). Trimmed reads from each sample were independently mapped to the *Drosophila melanogaster* genome release 6.19 using HISAT2 (*Kim et al., 2015*). Mapped reads were manipulated using *samtools* (*Li et al., 2009*).

### Statistical analyses, identification of DE genes and enrichment analyses

Read counts for each annotated gene were performed using htseq-count (*Anders et al., 2015*), where reads are counted at the exon level (using release 6.19 annotations obtained from the ENSEMBL Biomart) and then summed across all exons within a single gene. Total read counts for each gene for the 12 samples were then used for differential gene expression analysis using the Bioconductor package edgeR (*Robinson et al., 2010*) in R (*R Development Core Team, 2016*). We first filtered read counts by expression and removed lowly expressed genes. Read count data were normalised across libraries and expression dispersion parameters calculated in edgeR using the entire dataset.

Subsequently, expression data was subsetted into three parts for separate analysis, i) genes that were expressed in both sexes (transcripts detected in at least one replicate library of each sex), ii) genes that were male-limited in expression (transcripts detected in at least one replicate library from males, but none of the female libraries), and iii) genes that were female-limited in expression (transcripts detected in at least one replicate library from females, but none of the male libraries) (*Figure 1—figure supplement 1B*).

We tested for differential gene expression between our experimental groups using the negative binomial models implemented in edgeR. For the shared gene dataset, we fitted a full model where expression of each transcript is a function of sex, diet and their interaction. The significance of each model term was tested using a specific contrast matrix. In order to obtain estimates of expression fold changes between the two diets for each sex, we further fitted models with diet as the sole model term separately to male and female data.

Gene ontogeny enrichment was performed using the Bioconductor package *clusterProfiler* (*Yu et al., 2012*). In order to assess whether genes that showed similar diet responses were regulated by common transcription factors we used the Bioconductor package *RcisTarget* (*Aibar et al., 2017*), which tests for enrichment of cis-regulatory motifs upstream of a given gene sets. In all analyses, we used a statistical significance threshold of 5% False Discovery Rate (FDR) (*Benjamini and Hochberg, 1995*). For the smaller sex-specific gene sets, we ran enrichment analyses on the sets of genes with significant diet responses, but also on the complete sets of sex-limited genes (irrespectively of their responses to diet). This was to be able to identify (and remove) enriched binding motifs that reflect general sex-specific regulation rather than diet responses.

We further compared our list of genes responding to diet (either via additive or interactive effects) to previous work that has examined transcriptomic responses to dietary restriction (*Ding et al., 2014*). For this, we used the R package *GeneOverlap* (*Shen and Sinai, 2018*) that implements a contingency table test (Fisher's exact test) to identify greater than expected overlap between gene lists. To compare our gene list to genes reported as significantly affected by perturbation of IIS/TOR signalling by *Graze et al. (2018)*, we had to reanalyse their data using our pipelines. This was required because their analysis was performed at the exon-level, while we assessed transcerciption at the gene-level. We downloaded all raw data from the SRA database (SRP137911). We aligned all reads to the same version of the *Drosophila* nuclear genome we used for our analyses, and obtained gene-specific expression patterns across all their samples. We then applied the same statistical framework to these data that we had used for our own analysis, assessing the effect of sex, IIS perturbation and their interaction to genes expressed in both sexes. Overlap between classifications based on diet- and IIS/TOR perturbation-responses were assessed with $X^2$ tests and only considered genes that showed male and female expression in both datasets (N=8310).

## Experiment 3: Fitness response to diet and rapamycin

Male and female flies were assayed for fitness in the same way as previously described for Experiment 2. However, rather than just feeding either a protein-rich or a carbohydrate-rich diet, we combined each of the two dietary treatments with one of four different concentrations of the drug rapamycin (0 µM, 5 µM, 10 µM, 50 µM). Rapamycin is a drug that very specifically inhibits TORC1, and hence TOR-signalling, with this function being highly conserved from *S. cerevisiae* to humans (*Crespo and Hall, 2002*). Nutritional compositions and rapamycin levels were combined in a full factorial design resulting in a total of eight different diets (two nutritional compositions times four rapamycin levels, eight diets in total) for each sex. We had approximately 20 vials for each experimental unit.

We performed a joint analysis on a dataset combining male and female fitness data. Before statistical analyses, male and female fitness measures were transformed to obtain normally distributed residual values. Female egg numbers were log-transformed, whereas male competitive fertility data was arcsine-transformed. Moreover, to be able to compare across sexes, male and female fitness measures were further centred and scaled (separately for each sex) using Z-transformations. We fitted a linear fixed effects model to the transformed fitness values with sex, diet and rapamycin concentrations (coded as a categorical factor to accommodate possible non-linearity in the effect) and their interactions. For the main analysis we categorised diet as optimal/non-optimal (where the nutritional composition of the 'optimal' diet category is carbohydrate-rich for males and protein-rich for females). This encoding makes it more straightforward to assess how rapamycin treatment interacts with diet-quality in each sex. We also ran analysis where diet composition was encoded as 'carbohydrate-rich' and 'protein-rich'.

## Acknowledgements
We would like to thank Nazif Alic and Adam Dobson for their help with planning the experiments, analysis, and for insightful comments on previous versions of this manuscript. In addition, we thank Kevin Fowler, Filip Ruzicka and Michael Jardine for useful discussions and Rebecca Finlay for help with laboratory work. This study was funded by a Marie Sklodowska-Curie Research Fellowship [#708362] to MFC.

## Additional information

### Funding

| Funder | Grant reference number | Author |
| --- | --- | --- |
| European Commission | #708362 | M Florencia Camus |

The funders had no role in study design, data collection and interpretation, or the decision to submit the work for publication.

### Author contributions
M Florencia Camus, Conceptualization, Data curation, Formal analysis, Funding acquisition, Validation, Investigation, Visualization, Methodology, Writing—original draft, Project administration, Writing—review and editing; Matthew DW Piper, Conceptualization, Validation, Methodology, Writing—review and editing; Max Reuter, Conceptualization, Formal analysis, Methodology, Writing—original draft, Writing—review and editing

### Author ORCIDs
M Florencia Camus https://orcid.org/0000-0003-0626-6865
Matthew DW Piper http://orcid.org/0000-0003-3245-7219
Max Reuter http://orcid.org/0000-0001-9554-0795

### Decision letter and Author response
Decision letter https://doi.org/10.7554/eLife.47262.028
Author response https://doi.org/10.7554/eLife.47262.029

## Additional files

### Supplementary files
• Supplementary file 1. This datafile contains synthetic media recipes (tables 1-3), statistical analysis for Experiment 3 (table 4), and Chi$^2$ analysis for overlaps (table 5).
DOI: https://doi.org/10.7554/eLife.47262.020

• Supplementary file 2. Gene lists. In each tab, we show the genes that were significant in our analyses (FDR < 0.05)
DOI: https://doi.org/10.7554/eLife.47262.021

• Supplementary file 3. Transcription factor enrichment analysis. For each gene category of genes, we searced for enriched transcription factor motifs. This was done by surveying 5 kb upstream of every gene for enriched motifs.
DOI: https://doi.org/10.7554/eLife.47262.022

• Supplementary file 4. Diet-dependent expression responses of TOR signalling components. The table shows male and female fold changes (from high-carbohydrate to high-protein diet) for sets of genes associated with TOR signalling. We used two different methods to identify such genes. Sheets labelled 'IIS TOR' contain genes with the Gene Ontology annotations 'insulin signalling' or 'TOR signalling'. Sheet 'IIS TOR (all)' shows the overlap between these genes and genes in our dataset (irrespective of significance of differential expression). Sheets 'IIS TOR (concordant)' and "IIS TOR (opposing) show genes with sexually concordant and opposing expression responses, respectively (again, irrespectively of significane). Genes in bold show significant expression responses

(FDR < 0.05). The sheet 'REPTOR (all)' lists genes in our dataset that overlap with the TOR-responsive gene set reported by *Tiebe et al. (2015)*. Again, male and female fold changes are shown, irrespecively of their significance.

DOI: https://doi.org/10.7554/eLife.47262.023

• Transparent reporting form

DOI: https://doi.org/10.7554/eLife.47262.024

## Data availability

All data generated or analysed during this study are included in the supplementary data files. We compare our results to data from three papers, with results discussed throughout our paper (Dobson et al., 2018; Graze et al., 2018; Tiebe et al., 2015).

The following dataset was generated:

| Author(s) | Year | Dataset title | Dataset URL | Database and Identifier |
|---|---|---|---|---|
| Camus F, Piper M, Reuter M | 2019 | Data from: Sex-specific transcriptomic responses to changes in the nutritional environment | https://dx.doi.org/10.5061/dryad.2q1301v | Dryad Digital Repository, 10.5061/dryad.2q1301v |

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
