## [Decision Letter]

Thank you for submitting your article "Sex-specific transcriptomic responses to changes in the nutritional environment" for consideration by *eLife*. Your article has been reviewed by two peer reviewers, and the evaluation has been overseen by a Reviewing Editor and Michael Eisen as the Senior Editor. The following individual involved in review of your submission has agreed to reveal their identity: Felix Zajitschek (Reviewer #2).

The reviewers have discussed the reviews with one another and the Reviewing Editor has drafted this decision to help you prepare a revised submission.

This paper provides a detailed report on sex-specific transcriptomic responses to different diets in *D. melanogaster*. The analyses show that males and females share a large core of metabolic genes that are respond in a similar was to diet and a smaller set of genes that exhibit sex-specific response to diet. The later genes are potentially very interesting in terms of understanding sex-specific dietary fitness optima. The authors also show that sex-specific dietary changes are mediated by the TOR pathway.

Both reviewers liked the paper. They make useful suggestions about how the manuscript can be improved. The changes include some additional analyses and a more thorough discussion of the data. I believe that the proposed changes can readily be done -and should be done.

*Reviewer #1:*

I enjoyed reading this paper. This is a well written and well performed study of an important evolutionary and physiological problem, namely the nature of sex-specific transcriptomic responses to different diets in the fruit fly *Drosophila melanogaster*. Examining the transcriptional underpinnings of dietary responses is very interesting because female and male flies are known to have distinct, i.e. sex-specific fitness, optima with regard to their dietary choice / intake. While this is well appreciated and has been well investigated, e.g. using the nutritional geometry framework, still very little is known about the detailed underlying (transcriptional or other) mechanisms. This study seeks to fill this important gap. The key findings are novel and interesting: the sexes share a large core of metabolic genes that are qualitatively concordantly regulated in response to dietary changes/composition, i.e. genes that do not show sex-specific responses to diet. Yet, a smaller set of genes – especially genes involved in reproduction – exhibit sex-specific, divergent and opposite regulation in response to dietary change. These genes are potentially very interesting in terms of understanding sex-specific dietary fitness optima (even though this would require direct functional validation in future work). A proportion of these inversely regulated genes seems to be involved in nutrient sensing, i.e. apparently mainly the TOR signaling pathway. In line with this inference, an additional experiment (no. 3) using the TOR inhibitor rapamycin suggests that suppressing TOR signaling has a disproportionately negative effect on the sexes dietary optima. Together, these observations thus tentatively suggest that the TOR pathway might play a major role in mediating sex-specific dietary changes and in determining the sex-specific dietary fitness optima.

The study is technically very well and competently performed; the experimental methods are sound and state-of-the-art; and the bioinformatic and statistical analyses look robust.

My main concern is that the claim that TOR signaling plays a major role needs a bit more substantiation and elaboration: Table 3 shows that there is indeed overlap at the level of transcripts between this study and a previous study using rapamycin. But the reader is given rather little detail and thus a few major questions remain open: what is the overlap between the DR and the rapamycin treatments itself? Can all genes responding to rapamycin be called (canonical) TOR pathway genes? (i.e., how specific is rapamycin in its molecular action)? How strongly significant (actual values / quantification rather < 0.05 threshold) were the overlaps? Is there a way to quantify overall how special / exceptional the overlap with rapamycin treatment is (e.g., the overlap with the DR treatment seems to be stronger)? It would also be good to have a list or table or a figure of TOR signaling components that have been identified in the overlap as well as an indication of the directionality and an estimation of the effect sizes of the transcriptional responses ascribed to this pathway. Are there also IIS components among the overlapping genes, given that IIS and TOR form a signaling network and given that the study by Graze et al., 2018, found that IIS plays a major role in sex-specific gene expression? What is the overlap with the Graze et al. data? Thus, while I believe the overlap with the TOR pathway, and while the rapamycin manipulation experiment (experiment 3) supports the TOR inference, I feel that the major conclusion – i.e. that the sex-specific reverse regulation occurs via the TOR pathway – needs to be backed up better, with some more quantitative analysis (beyond the overlap and experiment 3). To my mind this is a rather important point but also one that should not be too difficult to address with the data already at hand.

*Reviewer #2:*

This is a very well designed, conducted, analysed and written up project, consisting of four separate experiments. It provides a thorough link between highly defined diet manipulation, sex-specific fitness measures, and gene expression analyses, with an additional in-depth experimental test of one sex-specific effects on reproductive fitness in one of the major nutrient-sensing pathways.

To identify sex-specific optimal diet, the authors used eight synthetic liquid diets that were isocaloric and differed only in their protein-to-carbohydrate ratio. Diet intake of 20 groups of three individuals, per sex per diet treatment, over four days, was associated with female fecundity and male competitive mating success. Diet preference was measured separately. Results of optimal diet assessment and diet preference supported previous findings, with females optimizing fecundity and preferring diet that is higher in protein (P:C=2:1), compared to males which optimized fertilization success and preferred higher carbohydrate diet (P:C=1:4).

For gene expression analyses, flies of each sex were provided with their optimal diet and the diet optimal for the opposite sex, in separate groups. Further expression analyses (differential expression, functional enrichment, upstream transcription factors) were performed on the genes that were expressed in both sexes, and genes that showed sex-limited expression. These analyses showed a large set of genes with sexually concordant regulation in response to protein-rich and carbohydrate-rich diets, and smaller sets of genes with either sexually antagonistic or sex-limited transcription patterns, which the authors then discuss in detail.

As an add-on, the authors also tested the effects of rapamycin on sex-specific fitness. Rapamycin is known for its inhibiting effect of the Tor pathway. Results showed a similar overall effect on male and female reproductive fitness, with a more pronounced effect in flies on their optimal diet, and a more dose-dependent effect in females, compared to inhibiting effects independent of the rapamycin dose in males.

---

## [Author Response]

Reviewer #1:

[…] *My main concern is that the claim that TOR signaling plays a major role needs a bit more substantiation and elaboration.*

Our interpretation that diet treatments elicit TOR-dependent responses is based on a combination of our bioinformatic analyses, showing overlap between diet-dependent genes and genes whose expression changes in response to DR, TOR-pathway manipulation, and direct experimental evidence of the diet-dependent effect of rapamycin treatment on male and female reproduction. While we believe that the experimental evidence is compelling, we revised the manuscript in line with this reviewer's comments to substantially bolster the bioinformatic support for the link between our dietary responses and TOR signalling (Introduction). Specifically, we now include an in-depth analysis of the Graze et al. dataset and the relationship between IIS/TOR and diet-dependent gene expression. This shows that the Graze et al. data show patterns of transcription that mirror those described in our data and significant overlap between IIS/TOR and diet responses (subsection “Statistical analyses, identification of DE genes and enrichment analyses”, last paragraph and subsection “Overlap with previously described diet and nutrient-signalling responses*”*). Furthermore, we show that genes responding to diet manipulation significantly overlap the best-characterised set of TOR-dependent genes currently available (Tiebe et al., 2015, subsection “Diet-specific regulation of male and female reproduction”).

We have edited the manuscript to incorporate these additional findings and clarify evidence for a link between diet responses and TOR signalling (Introduction).

Table 3 shows that there is indeed overlap at the level of transcripts between this study and a previous study using rapamycin. But the reader is given rather little detail and thus a few major questions remain open: what is the overlap between the DR and the rapamycin treatments itself?

The overlap between DR and rapamycin treatment has been previously described in Dobson et al., 2018. The authors found significant overlap between the genes whose expression changes significantly in response to DR and rapamycin treatment across 4 of the 5 female tissues samples. The two tissues that had the highest number of gene overlaps was the brain and the fat body, with 45 and 46 overlapping genes respectively. A summary of these findings can be found in Table 2 of Dobson et al., 2018. We have made the link between DR and rapamycin treatment clearer in the Introduction of our manuscript (fifth paragraph).

Can all genes responding to rapamycin be called (canonical) TOR pathway genes? (i.e., how specific is rapamycin in its molecular action)?

There has been a plethora of work investigating the specificity of rapamycin at inhibiting the TORC1 complex (see e.g. Crespo and Hall, 2002). Rapamycin is very specific in its molecular function, and this specificity is conserved across a wide range of taxa (from yeasts to humans). Accordingly, rapamycin treatment is considered an effective and specific means of manipulating TOR-dependent genes. We have added text to our Materials and methods section to clarify this point.

How strongly significant (actual values / quantification rather < 0.05 threshold) were the overlaps?

We have added p-values for all overlaps in Table 3.

Is there a way to quantify overall how special / exceptional the overlap with rapamycin treatment is (e.g., the overlap with the DR treatment seems to be stronger)?

We agree that the overlap between DR-responsive genes in Dobson et al. and diet-responsive genes in our study appears greater than that of rapamycin-responsive genes. On the other hand, the number of genes in the DR set are larger than those in the rapamycin set for most tissues (meaning that overlap might actually be greater, proportionally, for the latter comparison). Significance, on the other hand, will favour comparisons of larger sets that are more powerful.

On balance, we believe that quantification would not be satisfactory. Yet, we also think that this point is less important in the revised version of our manuscript, that is couched more generally in terms of IIS/TOR signalling. We hope the reviewer will agree with this assessment.

It would also be good to have a list or table or a figure of TOR signaling components that have been identified in the overlap as well as an indication of the directionality and an estimation of the effect sizes of the transcriptional responses ascribed to this pathway.

We have added a supplementary table as requested (Supplementary file 4). The identification of “TOR signalling components” was performed in two different ways. We first compiled a list of genes that can be found under the GO terms “TOR signalling” and “insulin signalling” (N=61 genes in total). We report fold changes in males and females for these genes, irrespectively of significance. This analysis gives an indication of directionality in expression for each sex, and is analogous to some of the analyses performed by Graze et al. [30] (see their Figure 6). We find that while some of the genes are concordant in their expression changes, a number of these genes, including components of TORC1 complex, show opposing expression responses.

For the second approach, we curated data from a study which we believe contains the most well-defined TOR-responsive gene set (Tiebe et al., 2015). These genes are regulated by REPTOR and REPTOR-BP and mediate most of the transcriptional induction caused by TORC1 inhibition in *Drosophila*. Although this work has only been carried out in males, it is currently the best characterisation of TORC1 effects in flies. We found a significant overlap between our set of genes and TORC1 responsive genes, and this analysis was included in our study (Table 3C, lines 624-631). Furthermore, to examine the directionality of these responses, we provide fold change in both males and females in Supplementary file 4.

Are there also IIS components among the overlapping genes, given that IIS and TOR form a signaling network and given that the study by Graze et al., 2018, found that IIS plays a major role in sex-specific gene expression? What is the overlap with the Graze et al. data?

As mentioned above, we have re-analysed the raw data of Graze et al. using our pipeline (to apply gene- not exon-level transcription levels, consistent with the rest of our analyses). Our re-analysis validates Graze et al.'s overall conclusions, with many genes being differentially expressed when InR is perturbed. However, it also shows that genes fall into classes corresponding to those we define based on diet composition, namely those where InR perturbation has a sexually concordant effect (class 'InR'), those where it has opposing effects in males and females ('InR×Sex'), and those where it elicits sex-biased/-specific effects (InR+'InR×Sex').

Comparing the genes affected by InR perturbation with those responding to diet treatments in our study, we find a significant overlap ate every level, including whether genes show any response in the two datasets, between the significance of individual terms (InR vs. D and InR×Sex vs. D×Sex) and between InR and diet-based gene classifications. We describe these results in the subsection “Overlap with previously described diet and nutrient-signalling responses” and discuss their implications in the subsection “Diet-specific regulation of male and female reproduction”. Furthermore, we have added a supplementary figure where we show the expression of these TORC1 responsive genes, indicating the strength and directionality of response across both sexes (Figure 2—figure supplement 1 and 2).

We have also included text in the Discussion (825-843) to address the important point made by the reviewer about TOR lying within a wider IIS/TOR network. We highlight the fact that while the observed phenotypes are TOR-dependent responses to rapamycin treatment, TOR itself participates in a broader set of (tissue specific) signalling responses and the diet responses observed in our main experiment could be due to regulation via TOR as well as IIS.

Thus, while I believe the overlap with the TOR pathway, and while the rapamycin manipulation experiment (experiment 3) supports the TOR inference, I feel that the major conclusion – i.e. that the sex-specific reverse regulation occurs via the TOR pathway – needs to be backed up better, with some more quantitative analysis (beyond the overlap and experiment 3). To my mind this is a rather important point but also one that should not be too difficult to address with the data already at hand.

We would like to thank the reviewer for their suggestions. We believe that the additional analyses have significantly strengthened our study. While establishing the link between diet responses and IIS/TOR signalling will ultimately require careful experimentation, the additional evidence we now include go, we believe, a long way to corroborating our inferences.